



# Time difference of arrival estimation of microseismic signals based on alpha-stable distribution

Rui-Sheng Jia[1,2], Yue Gong[1,2], Yan-Jun Peng[1,2], Hong-Mei Sun[1,2], Xing-Li Zhang[1,2], Xin-Ming Lu[1,2]

[1] College of Computer Science and Engineering, Shandong University of Science and Technology, Qingdao 266590, P.R. China;

[2] Shandong Province Key Laboratory of Wisdom Mine Information Technology, Shandong University of Science and Technology, Qingdao 266590, P.R. China;

*Correspondence to*: Rui-Sheng Jia (jrs716@163.com)

**Abstract.** Microseismic signals are generally considered to follow the Gauss distribution. A comparison of the dynamic characteristics of sample variance and the symmetry of microseismic signals with the signals which follows α-stable distribution, reveals that the pulse characteristics of the microseismic signal is outstanding and that the probability density curve of the microseismic signal is approximately symmetric. Thus, the hypothesis that microseismic signals follow the symmetric α-stable distribution is proposed. On the premise of this hypothesis, the characteristic exponent α of the seismic signals is obtained by utilizing the fractional low-order statistics, and then a new method of time difference of arrival (TDOA) estimation of microseismic signals based on fractional low-order covariance (FLOC) is proposed. Upon applying this method to the TDOA estimation of Ricker wavelet simulation signals and real microseismic signals, experimental results show that the FLOC method, which is based on the assumption of the symmetric α-stable distribution, leads to enhanced spatial resolution of the TDOA estimation relative to the generalized cross correlation (GCC) method, which is based on the assumption of the Gaussian distribution.

## 1. Introduction

Microseismic monitoring technology has been widely applied to mine rock burst monitoring, oil and gas field fracturing monitoring, reservoir seismic monitoring, slope stability evaluation and so on. Seismic source location is one of the key technologies used. The conventional seismic source localization method usually needs to pick up the P-arrival time of multi-channel seismic signals at first and then calculate the TDOA of the signals to solve the equation to obtain the source location. As a result, the accuracy of the calculated TDOA directly affects the accuracy of the seismic source location. However, in the process of actual operation, the first arrival time of the microseismic signals is not obvious, and there is much external noise. Therefore, it becomes very difficult to determine the time difference between waves from the same seismic source.

The basic problem the TDOA solves is to measure and estimate the TDOA between waves from the same seismic

source accurately and rapidly. Since the classic article on TDOA written by Knapp and Carter was published in 1976, this

problem has always been a research focus in the field of international signal processing (Knapp et al., 1976). The common

method of TDOA includes the generalized cross correlation method (Knapp et al., 1976; Souden et al., 2010; Jin et al., 2013),

the phase spectrum estimation method (Youn et al., 1982; Qiu et al ., 2012), the generalized bispectral estimation method

(Hinich et al., 1992; Hou et al., 2013), the adaptive estimation method (Gedalyahu et al., 2010; Salvati et al., 2013), the

energy method based on Hilbert-Huang transform (Sun et al., 2016) and so on. These methods have been widely used in

many fields. However, the vast majority of these methods assume that the signals and noises follow the Gaussian distribution.

In the case of non-Gaussian distributions, their algorithm has a serious degradation in spatial resolution and does not

function anymore (Ma et al., 1996; Cornelis et al., 2010; Park et al., 2011).

Noisy microseismic signals have conspicuous non-stationary characteristics, such as impulsiveness and randomness;

therefore, they belong to the category of non-Gaussian signals. If the microseismic signal is simulated by Gaussian signal, it

is inevitable that the TDOA algorithm will have a serious degradation in performance. To solve the problem in theory, we

intend to introduce the $\alpha$-stable distribution to describe the microseismic signal and noise in the distribution model. The

$\alpha$-stable distribution model has achieved excellence in the field of non-Gaussian signal processing, such as seismic inversion,

speech denoising and enhancement, sound source localization and mechanical fault diagnosis (Li et al., 2010; Yue et al.,

2012; Zhang et al., 2014). However, its use is rare in research projects and publications on TDOA estimation of microseismic

signals from the same seismic source. This paper fits the microseismic signal and noise as the signals to $\alpha$-stable distributions,

and studies the impact of non-Gaussian noise on the spatial resolution of TDOA, and proposes an improved TDOA algorithm

based on FLOC. Compared with the traditional TDOA algorithm, this improved algorithm could inhibit both the Gaussian

noise and the $\alpha$-stable distribution noise.

**2. The TDOA Model**

**2.1. The Basic Model of TDOA**

The basic model of TDOA is shown in the figure (Fig. 1). $s(n)$ represents the original microseismic signal. It spreads to the

two seismic geophones $S_1$ and $S_2$ through the rock stratum. Due to different propagation paths, the time at which the signals

arrive at the geophones are different.

(Figure 1 is about here)

If the microseismic acquisition system is discrete, the signals received by the geophones $S_1$ and $S_2$ can be expressed as:

$$\begin{cases} x_1(n) = \lambda_1 s(n) + b_1(n) \\ x_2(n) = \lambda_2 s(n-D) + b_2(n) \end{cases},$$   (1)

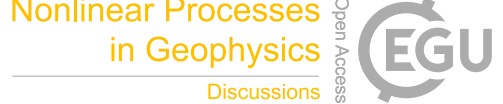



In the above equation, $s(n)$ represents the original signal. $D$ is the time delay value. $b_1(n)$ and $b_2(n)$ are the external Gaussian noises. In addition, $s(n)$, $b_1(n)$ and $b_2(n)$ are uncorrelated.

Correlation analysis is commonly used to calculate the TDOA estimation of two signals. In the case that the substance of the problem is not affected and the calculation is simplified, we take $\lambda_1=\lambda_2=1$. Then, the cross correlation function of the two microseismic signals $x_1(n)$ and $x_2(n)$ can be represented as:

$$
\begin{aligned}
R_{x_1 x_2}(\tau) &= E\left[x_1(n)x_2(n+\tau)\right] \\
&= R_{ss}(\tau-D) + R_{sb_1}(\tau-D) + R_{sb_2}(\tau) + R_{b_1 b_2}(\tau)
\end{aligned}
\tag{2}
$$

where $R_{ss}(\cdot)$ represents the auto-correlation function of the original signal. $R_{pq}(\cdot)$ is the cross-correlation function of the two

signals $p$ and $q$. It is assumed that $s(n)$, $b_1(n)$ and $b_2(n)$ are unrelated Gaussian noises. Then:

$$
R_{sb_1}(\tau-D) = R_{sb_2}(\tau) = R_{b_1 b_2}(\tau) = 0,
\tag{3}
$$

Eq. (2) can be rewritten as

$$
R_{x_1 x_2}(\tau) = R_{ss}(\tau-D),
\tag{4}
$$

The auto-correlation function has the property

$$
\left|R_{ss}(\tau-D)\right| \le R_{ss}(0).
\tag{5}
$$

Therefore, $R_{ss}$ is maximized when $\tau-D=0$. Thus, the TDOA estimation between the two seismic geophones can be expressed by the maximum of $R_{ss}(\tau-D)$.

$$
\hat{D} = \arg\left\{\max_{\tau}\left[R_{x_1 x_2}(\tau-D)\right]\right\}.
\tag{6}
$$

When noisy signal follows the Gaussian distribution, the above method can estimate the time delay by detecting the

peak position of the cross-correlation function of the signals $x_1(n)$ and $x_2(n)$. Based on this observation, the authors propose the generalized cross correlation method (Hertz et al., 1985; Kang et al., 2012; Zhang et al., 2015), the phase spectrum estimation method (Harada, 2014; Choudhuri et al., 2016), the adaptive estimation method (Carrier and Got, 2014; Wang et al., 2017) and so on to implement the TDOA estimation, which significantly improves the anti-noise property, estimation accuracy and resolution of the algorithm. However, these algorithms are all based on the second-order statistics and the

assumption that the noises follow the Gaussian distribution. In the process of microseismic monitoring, Noisy signals are non-Gaussian, and their pulse characteristics are obvious (Xu et al., 2015; Jia et al., 2016). therefore, Eq. (3) is not suitable for them. Therefore, it's necessary to introduce other models to describe the distribution characteristics of the microseismic signals and noises, and establish a new method to estimate the time delay.





### 2.2. The Model of α-stable Distribution

In the process of microseismic monitoring, external noises are composed of man-made noises, mechanical vibration, etc. The

common characteristics of these noises are that their time domain waveforms have conspicuous pulse characteristics, the

energy diminishes from low to high frequencies, and their corresponding probability density functions have a thicker tail

than that of Gaussian signals. In the field of signal processing, this type of non-Gaussian noise is usually described by the

α-stable distribution model.

The α-stable distribution is a random signal model that can be applied to an extensive range of problems. Except for a

few specific situations, there is no uniform probability density function expression; therefore, the following Eigen function is

used to express it (Shao and Nikias, 1993).

$$\varphi(\mathrm{t}) = \exp\left\{ \mathrm{j}\delta t - \gamma |t|^{\alpha} \left[ 1 + \mathrm{j}\beta \operatorname{sgn}(t)\omega(t,\alpha) \right] \right\}, \tag{7}$$

And

$$\omega(t,\alpha) = \begin{cases} \tan\dfrac{\alpha\pi}{2}, \alpha \neq 1 \\ \dfrac{\pi}{2}\log|t|, \alpha = 1 \end{cases}, \tag{8}$$

$$\operatorname{sgn}(t) = \begin{cases} 1, t > 0 \\ 0, t = 0 \\ -1, t < 0 \end{cases}, \tag{9}$$

where $\alpha$ represents the characteristic exponent. Smaller the values of $\alpha$ result in thicker tails of the probability density

function. $\beta$ is the skew parameter, representing the deviation degree of signals. It is a symmetric α-stable distribution signal

when $\beta$=0, which is also called the *SαS* distribution. $\gamma$ is the scale parameter, representing the dispersion degree of signal

around the location parameters, which is similar to the variance in the Gaussian distribution. $\delta$ is the location parameter,

which is similar to the mean or mid-value in the Gaussian distribution.

We can infer from Eq. (7) and (8) that the corresponding Eigen function is the same as when $\alpha$=2. That is to say, the

Gaussian distribution is a special case of the α-stable distribution. When $0<\alpha<2$, the Eq. (7) represents the Eigen function of

the signals following the non-Gaussian distribution, which is also called the fractional lower-order α-stable distribution.

### 2.3. Non-Gaussian Property of Microseismic Signals

The difference of judging a signal between the Gaussian distribution and the α-stable distribution is that the latter has

stronger pulse characteristics. Due to the existence of the pulse, the secondary moment of the observation data that follow

the $\alpha$-stable distribution is not convergent, and there is no limited high-order moment above the second order. However, the

observation data that follow the Gaussian distribution have both stable secondary moment and limited high-order moment



(Sun and Qiu, 2008). Therefore, whether the signal follows the Gaussian distribution can be judged from whether the

sample variance of the observed data is convergent.

If $x_i$, $i=1,2,3,…N$ represents the observed data sequence and $N$ represents the sample number of observed data, the

dynamic sample variance of the first $k \left(1 \le k \le N\right)$ observed data is defined as

$$S_k^2 = \frac{1}{k} \sum_{i=1}^{k} \left(x_i - \bar{x}\right)^2 \ , \tag{10}$$

And

$$\bar{x} = \frac{1}{k} \sum_{i=1}^{k} x_i. \tag{11}$$

With the continuous increase of $k$, if $S_k^2$ converges to a certain value, the observed data sequence follows the Gaussian

distribution. Otherwise, it follows the α-stable distribution. To illustrate the changes of the dynamic sample variance of the

Gaussian signals and the α-stable distribution signals, three sets of random data are produced for comparison. The sample

length of the three sets of data are all 1000 (Fig. 2), and the figure (Fig. 2a) is a Gaussian signal. This means that $α=2.0, β=0$,

$γ=1, δ=0$;The figure (Fig. 2b) is a random signal that follows the α-stable distribution, and $α=1.6, β=0, γ=1, δ=0$;The figure

(Fig. 2c) is another random signal following the α-stable distribution, and $α=1.2, β=0, γ=1, δ=0$;The figures (Fig. 2a*, Fig.

2b*,Fig. 2c*) are the dynamic sample variances corresponding to signals (Fig. 2a, Fig. 2b, Fig. 2c), respectively.

(Figure 2 is about here)

A comparison of the waveform characteristic of the signals (Fig. 2a, Fig. 2b, Fig.2c) shows that with the gradual

decrease of the characteristic exponent α, the pulse characteristic of signals is enhanced. The signal (Fig. 2a) follows the

Gaussian distribution. Its pulse characteristic is not obvious, and its dynamic sample variance converges to a stable value.

The characteristic exponent $α$ of the signal (Fig. 2b) is 1.6. It has a strong pulse characteristic. Its dynamic sample variance

springs stepwise and does not converge to a stable value with an increase in sample points. The characteristic exponent $α$ of

the signal (Fig. 2c) is 1.2. Its pulse characteristic is more obvious. The step amplitude of the dynamic sample variance

increases sharply; therefore, it is more difficult to converge to a stable value.

We select a measured microseismic wave and calculate its dynamic sample variance according to Eq. (10) (Fig. 3), It

shows that the microseismic signal's dynamic sample variance jumps stepwise and does not converge to a stable value either.

Thus, one can conclude that the microseismic signal follows the fractional lower-order α-stable distribution. Through the

analysis of a large number of seismic signals and the calculation of characteristic exponents, current literature (Yue et al.,

2013) shows that the characteristic exponent $α$ of a seismic signal is less than 2, usually between 1.8458 and 1.9301.

(Figure 3 is about here)

**2.4. The Judgment of Symmetry Property of Microseismic Signal**

Before the parameter estimation of the α-stable distribution, we should determine whether the distribution of the signal is

symmetric The methods for identifying symmetry are listed below:

(1)    Draw the probability density curve of the sample sequence and observe the symmetry

(2)    Count the number of positive and negative values in the sample sequence. If the number of positive and negative

values are approximately same, the signal is symmetric.

The figure (Fig. 4a) shows that when the skew parameter $\beta=0$, the probability density curve is symmetric; when $\beta=0.8$,

the probability density curve is right-skewed; and when $\beta=-0.8$, the probability density curve is left-skewed. The figure (Fig.

4b) shows 5 probability density curves of microseismic signals from the same seismic source. It is obvious that these curves

are symmetric. As a result, the distribution of microseismic signal is considered symmetric.

(Figure 4 is about here)

For further validation of the symmetry of microseismic signal, we randomly select 30 signals from the microseismic

records in different places, truncate the continuous 3000 sampling points of each signal and then count the numbers of

positive and negative. The absolute value for the difference between the numbers of positive and negative is shown in the

figure (Fig. 5).

(Figure 5 is about here)

The figure (Fig. 5) shows that the maximum difference between the number of positive and negative values is 92.

Compared with the 3000 of sample data, this can be approximately considered as 0. Therefore, the microseismic signal is

approximately considered symmetric.

In conclusion, the microseismic signal follows the symmetric α-stable distribution, which is also called the *SαS*

distribution. Because the α-stable distribution does not have limited secondary and high-order moment, the above TDOA

method is based on the assumption that the secondary moment   (or high-order moment) and the Gaussian noise has a

serious degradation in performance. It is necessary to do some research on the new TDOA algorithm based on the low-order

statistics.

**3. The TDOA Estimation Algorithm**

**3.1. The TDOA Estimation Based on FLOC**

In the case that the noise follows the α-stable distribution, an existing study (Ma and Nikias, 1995) puts forward a TDOA

algorithm based on fractional low-order covariance (FLOC). The FLOC of two signals $x_i(t)$ ($i=1,2$) is defined as:





$$R_d(\tau) = E\left[x_2(t)^{<A>} x_1(t+\tau)^{<B>}\right], 0 \le A < \frac{\alpha}{2}, 0 \le B < \frac{\alpha}{2}, 0 < \alpha \le 2, \qquad (12)$$

and

$$x^{<c>} = |x|^c \operatorname{sgn}(x), \operatorname{sgn}(x) = \begin{cases} 1, x > 0 \\ 0, x = 0 \\ -1, x < 0 \end{cases}, \qquad (13)$$

where $A$ and $B$ represent the fractional low-order exponents of the two input signals $x_i(t)$ ($i$=1,2), respectively. $\tau$ is the translation relative to the signal $x_1(t)$ when calculating FLOC. The TDOA estimation can be obtained by detecting the peak of the function $R_d(\tau)$.

$$D = -\arg\left\{\max_\tau\left[R_d(\tau)\right]\right\}. \qquad (14)$$

The FLOC algorithm can be used for the TDOA estimation of microseismic signals. If the two microseismic signal samples are $x_i(n)$ ($i$=1,2;$n$=1,2,…,$N$),the Eq. (12) can be expressed by

$$\hat{R}_d(\tau) = \frac{1}{N}\sum_{n=1}^{N}|x_2(n)|^A |x_1(n+\tau)|^B \cdot \operatorname{sgn}\left[x_2(n)x_1(n+\tau)\right],$$
$$0 \le A < \frac{\alpha}{2}, 0 \le B < \frac{\alpha}{2}, 0 < \alpha \le 2, \qquad (15)$$

The TDOA estimation can be obtained by detecting the peak of the function $\hat{R}_d(\tau)$.

$$\hat{D} = -\arg\left\{\max_\tau\left[\hat{R}_d(\tau)\right]\right\}. \qquad (16)$$

The TDOA method based on FLOC require very few calculations, and has a strong real-time, simple implementation. However, the $\alpha$ parameter needs to be estimated in advance; otherwise, the FLOC algorithm will have a serious degradation in performance and will lead to incorrect results when $A$ and $B$ are greater than $\alpha/2$.

### 3.2. The Estimation of the Characteristic Exponent $\alpha$

For the random variable $X$, which follows the $\alpha$-stable distribution, the fractional lower-order moment is defined as $E\left(|X|^p\right), 0 < p < \alpha \le 2$. $p$ is the order of fractional lower-order moment. What we obtain from the Zolotarev theorem (Zolotarev, 1966) is that

$$E\left(|X|^p\right) = C(p,\alpha)\gamma^{p/\alpha}, \qquad (17)$$

and

$$C(p,\alpha) = \frac{2^p \Gamma\left(\frac{p+1}{2}\right)\Gamma\left(1-\frac{p}{\alpha}\right)}{\sqrt{\pi}\,\Gamma\left(1-\frac{p}{2}\right)}, \qquad (18)$$

where $\alpha$ represents the characteristic exponent, $\gamma$ represents the scale parameter, and $\Gamma(\cdot)$ represents the Gamma function.



If the random variable $X$ follows the $S\alpha S$ distribution, a study has found that there is a negative-order moment in the

$S\alpha S$ distribution (Ma and Nikias,1995). The Eq. (17) can then be changed to

$$E\left(|X|^p\right) = C(p,\alpha)\gamma^{p/\alpha}, -1 < p < \alpha \le 2, \tag{19}$$

because

$$E\left(|X|^p\right) = E\left(e^{p\log|X|}\right) = C(p,\alpha)\gamma^{p/\alpha}, \tag{20}$$

Eq. (20) is continuous at the point $p=0$ after the introduction of negative-order moment. If $Y = \log|X|$, $E\left(e^{pY}\right)$ is the

moment-generating function of $Y$ and

$$E\left(e^{pY}\right) = \sum_{k=0}^{\infty} E\left(Y^k\right)\frac{p^k}{k!} = C(p,\alpha)\gamma^{p/\alpha}, \tag{21}$$

Then, any order moments of $Y$ are limited and

$$E\left(Y^k\right) = \frac{d^k}{dp^k}\left[C(p,\alpha)\gamma^{p/\alpha}\right]\Big|_{p=0}, \tag{22}$$

This can be simplified to

$$E(Y) = C_e\left(\frac{1}{\alpha}-1\right) + \frac{1}{\alpha}\log\gamma, \tag{23}$$

where $C_e = 0.57721566...$, a Euler constant. Then

$$Var(Y) = E\left\{\left[Y - E(Y)\right]^2\right\} = \frac{\pi^2}{6}\left(\frac{1}{\alpha^2} + \frac{1}{2}\right). \tag{24}$$

For the microseismic signals $Y_i$ ($i$=1,2,…,$N$, $N$ is the sampling number), the mean and variance can be obtained by Eq.s

(25) and (26), respectively.

$$\bar{Y} = \frac{1}{N}\sum_{i=1}^{N} Y_i, \tag{25}$$

$$Var(Y) = \frac{1}{N}\sum_{i=1}^{N}\left(Y_i - \bar{Y}\right)^2. \tag{26}$$

Plugging in the value gained from the Eq. (26) into the Eq. (24), we can obtain an estimated value of $\alpha$. Then, the value

of $\alpha$ is plugged in Eq. (23), and we obtain the value of $\gamma$.

### 3.3. Algorithm Procedures

If $x_i(n)$($i$=1,2;$n$=1,2,…,$N$) represents the sample of two microseismic signals from the same seismic source, the TDOA

algorithm is shown below:

Step 1:For a given sequence of discrete signal $x_1(n)$ and $x_2(n)$, calculate their characteristic exponents $\alpha_1$ and $\alpha_2$

according to Eq.s (24), (25) and (26);

Step 2:Assigning $A = \frac{0.95 \times \alpha_1}{2}, B = \frac{0.95 \times \alpha_2}{2}$, we can easily know that $0 \le A < \frac{\alpha_1}{2}, 0 \le B < \frac{\alpha_2}{2}$ ;



Step 3:Add the Hanning window to $x_1(n)$ and $x_2(n)$, and set the window lengths to $max(size(x_1(n)))$ and $max(size(x_2(n)))$.

The cross-correlation function $\hat{R}_d(\tau)$ of $x_1(n)$ and $x_2(n)$ is calculated according to Eq. (15);

Step 4:Detect the peak of the function $\hat{R}_d(\tau)$. Then, the TDOA estimation $\hat{D}$ can be obtained.

**4. Simulation And Analysis**

The signals Ricker1 and Ricker2 used in the simulation are two Ricker wavelets. Their spectral peak frequency is 25 Hz.

The sampling frequency is 1 KHz, and the number of sampling points is 1000. The time delay between the two Ricker

wavelets is set to 70 ms (Fig. 6a). The generalized signal-to-noise ratio (GSNR) is defined in Eq. (27) and used to describe

the power ratio of signal and noise (Ma and Nikias, 1996) .

$$GSNR = 10\lg\frac{\sigma_s}{\gamma},\tag{27}$$

In the equation, $\sigma_s$ represents the signal power, and $\gamma$ represents the noise figure of the α-stable distribution.

(Figure 6 is about here)

**Experiment 1:** The spatial resolution on TDOA estimation of the GCC method based on the Gaussian distribution and

the FLOC method based on the non-Gaussian distribution, are compared and and verified. α-stable distribution noises to the

two Ricker wavelets are added. The parameter of the α-stable distribution ($\alpha$, $\beta$, $\gamma$, $\delta$) is set to (1.2, 0, 1, 0). Because of the

randomness of α-stable distribution noises, the two noises are independent of each other. The two Ricker wavelets with

added noises are shown in the figure (Fig. 6b). In the case of α-stable distribution noises, the TDOA estimation results of the

GCC method and the FLOC method when $GSNR$=0 dB are shown in the figures (Fig. 6c, Fig. 6d).

It is evident from the figures (Fig. 6c, Fig. 6d) that the GCC method has a serious degradation in performance when

there are α-stable distribution noises and $GSNR$=0 dB. There are several peak positions in the curve so that the correct result

is difficult to get. However, the FLOC method has a good anti-interference capability. The peak appears at 70 ms and can

estimate the time delay correctly.

**Experiment 2:** The influence of different $\alpha$ to the TDOA estimation results are verified. The two noises are generated

randomly when $\alpha$ takes different values between 0 and 2 and are added to Ricker1 and Ricker2, respectively. When $GSNR$=0

dB, the TDOA estimation result of the two signals with noises obtained by the FLOC method is shown in the figure (Fig. 7).

The figures (Fig. 7a, Fig. 7b, Fig. 7c, Fig. 7d) show the waveforms of Ricker1 and Ricker2 with different noise signals added

to these waveforms. The figures (Fig. 7a[*], Fig. 7b[*], Fig. 7c[*], Fig. 7d[*]) show the corresponding TDOA estimation results.

(Figure 7 is about here)

It is evident from the figure (Fig. 7) that a smaller $\alpha$ (which implies that there is a stronger pulse of noise) corresponds

to a better performance on the TDOA estimation of the FLOC algorithm. When $\alpha$ is close or equal to 2, the performance has a degradation (Fig. 7d*), but it still can obtain the correct TDOA estimation result: 70ms. Therefore, the FLOC method

performs well irrespective of the noise and follows the Gaussian distribution or the α-stable distribution.

## 5. CASE STUDY

To verify the effectiveness of the FLOC method for TDOA estimation of real microseismic signals, we select eight microseismic signals from the same seismic source to do the experiment. The eight signals come from the ISS microseismic monitoring system of a coal mine in central China. Seismic geophones are laid along the mining roadway every 50 m in the

system. The frequency bandwidth of the seismic geophones is between 3 Hz and 2000 Hz. The data acquisition frequency is 1 KHz. For convenience of comparing and analyzing the experiment results, the first 2000 sampling points of each waveform are picked as the data object. The P-arrival time of each microseismic signal are recorded manually, and the time delay between any two of the microseismic signals as a reference of the experimental result is calculated.

As an example, the microseismic signals in 2# and 7# roadway are selected to explain the result. The waveforms of the

microseismic signals after interception are shown in the figures (Fig. 8a, Fig. 8b). The time delay between the two signals obtained by manual method is 19ms. First, when the microseismic signal follows the Gaussian distribution, the PHAT-GCC method, which, of the GCC methods, performs best, is chosen for the TDOA estimation of microseismic signals from the same seismic source. The result is shown in the figure (Fig. 8c). Second, when the microseismic signal follows the α-stable distribution, FLOC is used for the TDOA estimation. The characteristic exponent $\alpha$ of the two picked signals are calculated

according to Eq.s (23), (24) and (25). $\alpha_2=1.802$, $\alpha_7=1.835$ are obtained as results. According to STEP 2 in section 3.3, assign $A$=0.85595 and $B$=0.87163. The TDOA estimation result obtained by the FLOC method is shown in the figure (Fig. 8d).

(Figure 8 is about here)

The figures (Fig. 8c, Fig. 8d) show that the two methods both obtain the correct result: 19 ms, but the peak of the FLOC method is sharper than the GCC method. This implies that the FLOC method performs better.

Each of the eight microseismic signals is considered to be a set of data following the α-stable distribution. Their characteristic exponent $\alpha$ are calculated according to Eq.s (23), (24) and (25) and are shown in the table (Table 1). The values are between 1.802 and 1.913. It can be seen that the characteristic exponent $\alpha$ of each signal are all less than 2. We can obtain 28 pairs of microseismic signals by the pair combination of the 8 signals. The comparison of TDOA estimations obtained by the PHAT-GCC, FLOC and manual method is shown in the table (Table 2).

(Table 1 is about here)

(Table 2 is about here)

An analysis of tables (Table 1, Table 2) indicates that the pulse of actual microseismic signal is stronger than the one following the Gaussian distribution. Because the characteristic exponent of the actual microseismic signal is less than 2, it is considered to be a signal following the α-stable distribution. Based on this observation, we can say that the spatial resolution

275        of the FLOC method is better than the PHAT-GCC method on the TDOA estimation of microseismic signals.

**6. Conclusions**

(1) Through the analysis of the convergence of dynamic sample variance, the microseismic signal with noises is proved to follow the α-stable distribution. The analysis of the symmetry of probability density curve of the sample sequence proves that the microseismic signal is approximately symmetric. Therefore, it is more reasonable to regard the microseismic signal

280        with noises as the α-stable distribution signal.

(2) Because of the absence of second-order statistics of α-stable distribution, one cannot obtain optimal or correct estimation values via the traditional TDOA method based on the Gaussian distribution.

(3) Microseismic monitoring data obtained from a coal mine in central China is used for TDOA estimation based on the GCC method and the FLOC method to study cases when the microseismic signals follow the Gaussian distribution and the α

285        stable distribution. In the comparison to the results and the time delay obtained manually, we observe that the FLOC method performs better than the traditional GCC method irrespective of whether the noise follows the Gaussian distribution or the α-stable distribution. This method is suitable for the TDOA estimation of microseismic signals from the same seismic source.

**Acknowledgments**

This work is funded by the State Key Research Development Program of China (2016YFC0801406), the Key Research and

290        Development Program of Shandong Province (2017GSF20115, 2016GSF120012), China Postdoctoral Science Foundation (2015M582117), Qingdao Postdoctoral Applied Research Project and Special Project Fund of Taishan Scholars of Shandong Province.

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





**TABLE**

**Table 1: The characteristic exponent $\alpha$ of microseismic signal**

| Roadway number | characteristic exponent $\alpha$ | Roadway number | characteristic exponent $\alpha$ |
|---|---|---|---|
| 1# | 1.864 | 5# | 1.913 |
| 2# | 1.802 | 6# | 1.857 |
| 3# | 1.822 | 7# | 1.835 |
| 4# | 1.901 | 8# | 1.846 |

**Table 2: The comparison of TDOA estimation results of microseismic signal**

| TDOA method | error=0ms percentage | error$\leqslant$3ms percentage | error$\leqslant$5ms percentage |
|---|---|---|---|
| FLOC | 96.43% | 100% | 100% |
| PHAT-GCC | 85.71% | 89.29% | 92.86% |



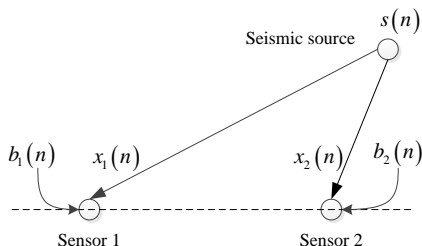

**Fig. 1: Two-sensor model of TDOA estimation**

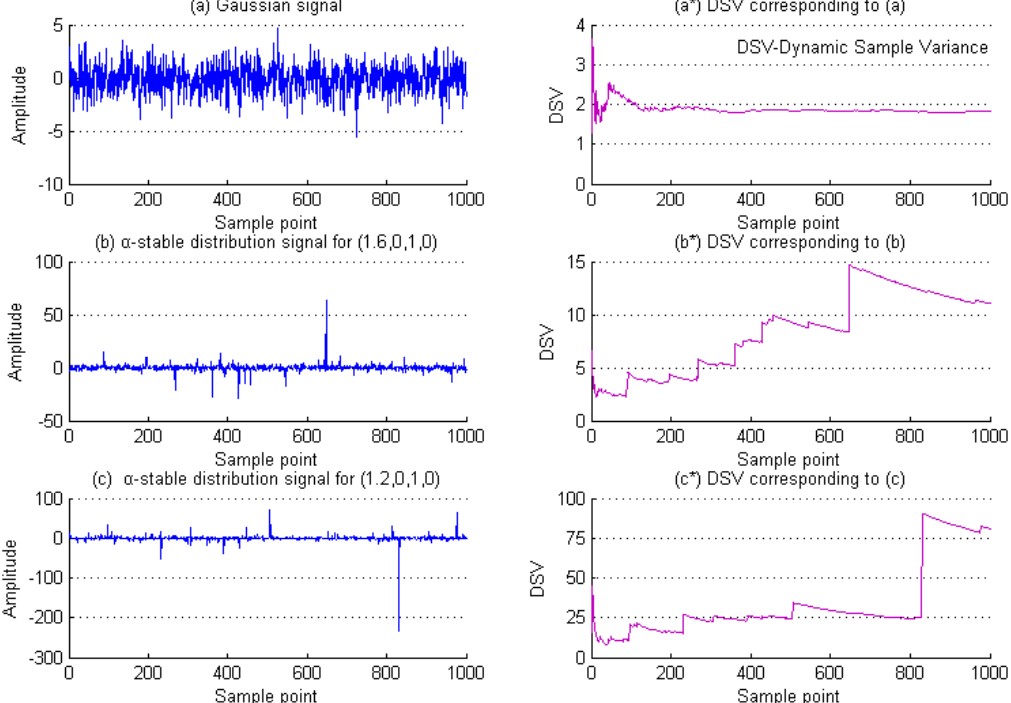

**Fig. 2: Gaussian signal, *α*-stable distribution signal and their dynamic sample variance**

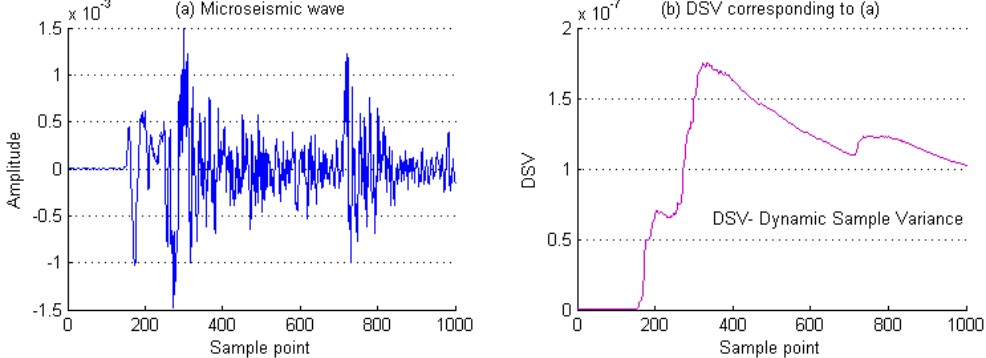

**Fig. 3: (a) Microseismic wave and (b) its dynamic sample variance**



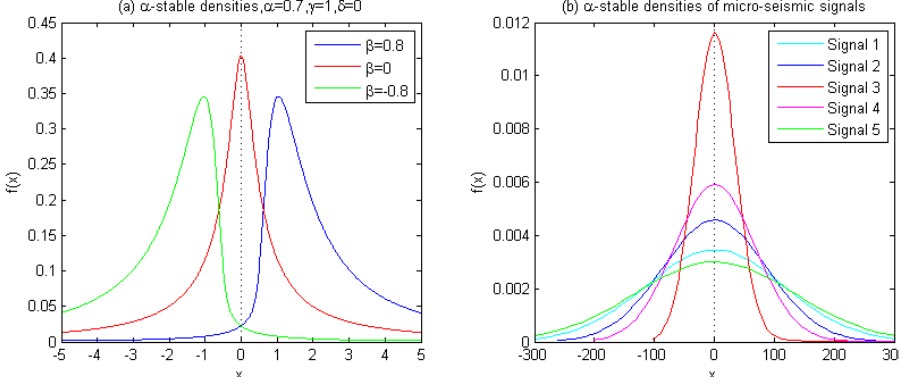

**Fig. 4: (a) The α-stable densities with different skew parameter; (b) The α-stable densities of microseismic signals.**

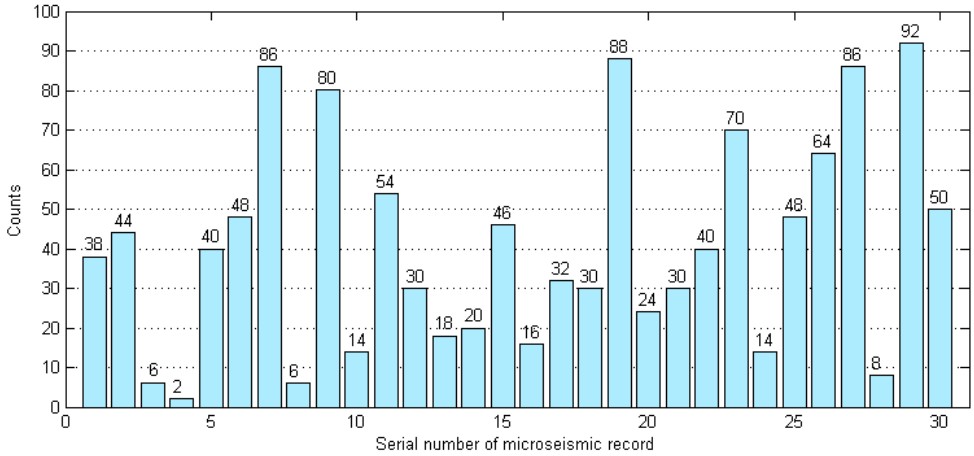


**Fig. 5: The absolute value for the difference between the positive and negative counts in each microseismic record**

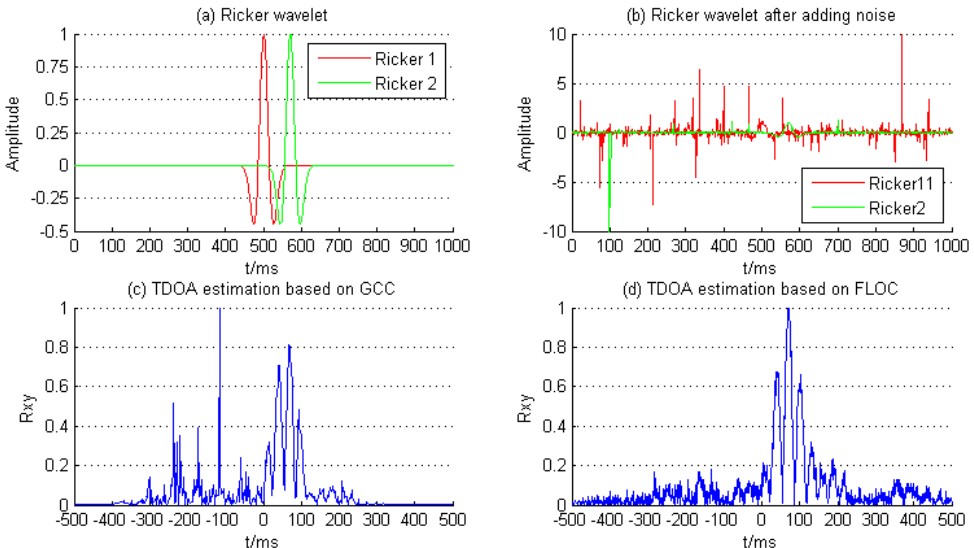

**Fig. 6: Comparison of the TDOA estimation results of GCC and FLOC method with α-stable distribution noise**



**Fig. 7: The influence of different α to the TDOA estimation result**

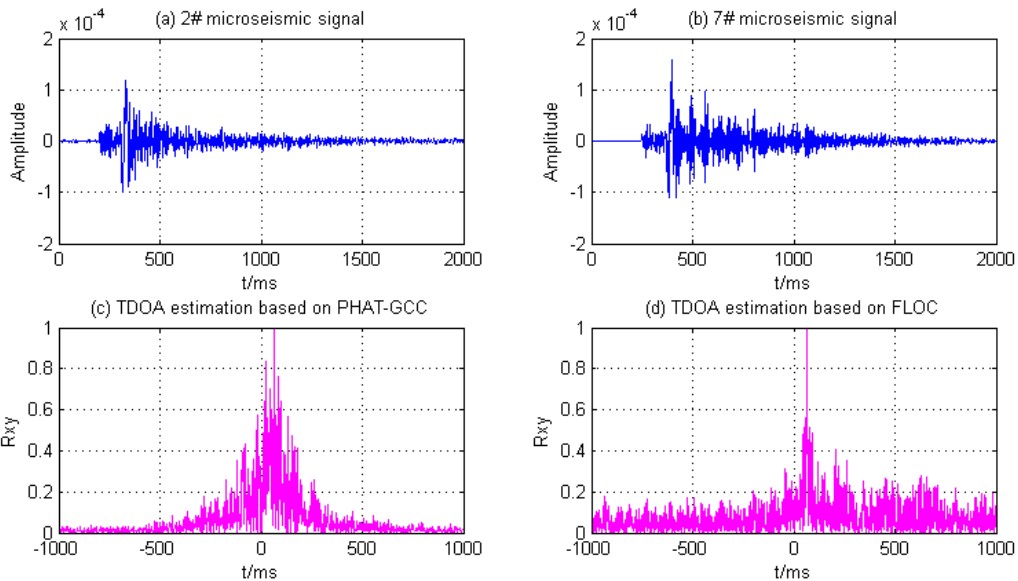

**Fig. 8: The comparison of TDOA estimation results based on PHAT-GCC and FLOC**