# Peer review of "Time difference of arrival estimation of microseismic signals based on alpha-stable distribution"

_Nonlinear Processes in Geophysics, 2017_

## Referee Comment (RC1) · Anonymous Referee #1 · 8 Jan 2018

After reviewing said manuscript, several points spring to mind with regards to improving paper

- A review of paper for grammatical errors and typos is suggested. Some missing punctuation in some places - Some clarity is required with regards to certain statements: e.g "The pulse characteristics of the microseismic signal is outstanding" seems vague. General contains a number of vague sentences - The examples provided haven't fully demonstrated how this is better than the generalized cross- correlation. - Paper isn't suitable for a general audience, provide better context as to why this is important/useful. Also provide some background to better understand concepts presented. - Figures are too small, try increasing font sizes on captions and figure size.

---

## Referee Comment (RC2) · Anonymous Referee #2 · 18 Jan 2018

Here are some suggestions, I wish these suggestions are useful to you. 1.The number of Eq.(7) is not aligned 2.Your innovation is not highlighted in the theoretical parts. 3.In step2 of 3.3, 0.95 should be introduced.  4.In experiment 1, the definition and formulas of GSNR should be introduced. 5.In Fig3(a), the energy of P wave is larger than the energy of S wave. But in most field microseismic data, the energy of S wave is larger than the energy of P wave, so I think your measured microseismic signal is not representative. 6.In Fig6, your method is just compared with one conventional method. I think this is not enough, your method should be compared with two conventional methods at least so that the advantages of your method are more persuasive. 7.In experimental part, your method is not tested by field microseismic data, you should add actual experiment in this paper.  8.Three components of microseismic signals should

be added in your paper. 9.In experimental part, the value of the GSNR is only one , I think you should change the value of GSNR and do more experiments. This can verify whether your method will be affected by the energy of the noises.

---

## Author Comment (AC1) · 21 Feb 2018

Referee #1

Q1. A review of paper for grammatical errors and typos is suggested. Some missing punctuation in some places - Some clarity is required with regards to certain statements: e.g "The pulse characteristics of the microseismic signal is outstanding" seems vague. General contains a number of vague sentences

A1: We are very sorry for our poor english writing. We have made correction according to the Reviewer's comments, see P1L11.

Q2. The examples provided haven't fully demonstrated how this is better than the generalized cross-correlation.

[Figure]

A2: It's really true as reviewer suggested that our metod is better than the generalized cross-correlation, we have added some experiments to verify the method by changing the value of GSNR, and we have added PHAT-GCC method to be compared. See P18.

Q3. Paper isn't suitable for a general audience, provide better context as to why this is important/useful. Also provide some background to better understand concepts presented.

A3: As reviewer suggested that paper isn't suitable for general audience,We have added a reference to introduced the background in paper. See P1L25.

Q4. Figures are too small, try increasing font sizes on captions and figure size.

A4: It's really true as reviewer suggested that figures are too small, We have increased fontsize on captions and figure size. See P15-P19.

Referee #2

Q1. The number of Eq. (7) is not aligned.

A1: We have made correction according to the Reviewer's comments. For the corretion detail, see P4L93.

Q2. Your innovation is not highlighted in the theoretical parts.

A2: We have re-written this part according to the Reviewer's suggestion. For the modified detail, see P6 L165-P7L169

Q3. In step2 of 3.3, 0.95 should be introduced.

A3: 0.95 is an empirical value, the purpose of which is to make the value of A, B satisfied , Considering the Reviewer's suggestion, we have introduced 0.95 in P9L220-221.

Q4. In experiment 1, the definition and formulas of GSNR should be introduced.

A4: We have introduced the definition and formulas of GSNR in P9L228-229.

Q5. In Fig3(a), the energy of P wave is larger than the energy of S wave. But in most field microseismic data, the energy of S wave is larger than the energy of P wave, so I think your measured microseismic signal is not representative.

A5: It is really true as Reviewer suggested that our measured microseismic signal is not representative, We have chosed a typical microseismic signal and redrew Fig. 3. For more detail , see Fig. 3 in P17.

Q6. In Fig. 6, your method is just compared with one conventional method. I think this is not enough, your method should be compared with two conventional methods at least so that the advantages of your method are more persuasive.

A6: Considering the Reviewer's suggestion, we have added PHAT-GCC method to be compared. See P18.

Q7. In experimental part, your method is not tested by field microseismic data, you should add actual experiment in this paper.

A7: As reviewer suggested that our method is not tested by field microseimic data in expriment part, we have added test by field microseismic data in section 5: Case Study. See P10.

Q8. Three components of microseismic signals should be added in your paper.

A8: We have added three component of microseismic siginals in the paper. See P2L52-53.

Q9. In experimental part, the value of the GSNR is only one , I think you should change the value of GSNR and do more experiments. This can verify whether your method will be affected by the energy of the noises.

A9: As reviewer suggested that we should change the value of GSNR and do more experiments, we have added some experiments to verify the method. See P18.

We tried our best to improve the manuscript and made some changes in the

manuscript. These changes will not influence the content and framework of the paper. And here we did not list the changes but marked in red in revised paper.

We appreciate for Reviewers' warm work earnestly, and hope that the correction will meet with approval. Once again, thank you very much for your comments and suggestions.

Please also note the supplement to this comment:
https://www.nonlin-processes-geophys-discuss.net/npg-2017-49/npg-2017-49-AC1-supplement.pdf

$s(n)$

Seismic source

$b_1(n)$    $x_1(n)$         $x_2(n)$    $b_2(n)$

Sensor 1                    Sensor 2

**Fig. 1.**

**Fig. 2.**

[Figure]

Interactive
comment

[Figure]

[Figure]

**Fig. 3.**

[Figure]

[Figure]

**Fig. 4.**

[Figure]

**Fig. 5.**

Fig. 6.

(a) α=0.8

(a*) Time delay estimation when α=0.8

(b) α=1.4

(b*) Time delay estimation when α=1.4

(c) α=1.8

(c*) Time delay estimation when α=1.8

(d) α=2.0

(d*) Time delay estimation when α=2.0

**Fig. 7.**

[Figure]

**Fig. 8.**

**Supplement:**

[revised manuscript text omitted]

---

## Author Comment (AC2) · 21 Feb 2018

Q1. The number of Eq. (7) is not aligned.

A1: We have made correction according to the Reviewer's comments. For the corretion detail, see P4L93.

Q2. Your innovation is not highlighted in the theoretical parts.

A2: We have re-written this part according to the Reviewer's suggestion. For the modified detail, see P6 L165-P7L169

Q3. In step2 of 3.3, 0.95 should be introduced.

A3: 0.95 is an empirical value, the purpose of which is to make the value of A, B

satisfied , Considering the Reviewer's suggestion, we have introduced 0.95 in P9L220-221.

Q4. In experiment 1, the definition and formulas of GSNR should be introduced.

A4: We have introduced the definition and formulas of GSNR in P9L228-229.

Q5. In Fig3(a), the energy of P wave is larger than the energy of S wave. But in most field microseismic data, the energy of S wave is larger than the energy of P wave, so I think your measured microseismic signal is not representative.

A5: It is really true as Reviewer suggested that our measured microseismic signal is not representative, We have chosed a typical microseismic signal and redrew Fig. 3. For more detail , see Fig. 3 in P17.

Q6. In Fig. 6, your method is just compared with one conventional method. I think this is not enough, your method should be compared with two conventional methods at least so that the advantages of your method are more persuasive.

A6: Considering the Reviewer's suggestion, we have added PHAT-GCC method to be compared. See P18.

Q7. In experimental part, your method is not tested by field microseismic data, you should add actual experiment in this paper.

A7: As reviewer suggested that our method is not tested by field microseimic data in expriment part, we have added test by field microseismic data in section 5: Case Study. See P10.

Q8. Three components of microseismic signals should be added in your paper.

A8: We have added three component of microseismic siginals in the paper. See P2L52-53.

Q9. In experimental part, the value of the GSNR is only one , I think you should change

the value of GSNR and do more experiments. This can verify whether your method will be affected by the energy of the noises.

A9: As reviewer suggested that we should change the value of GSNR and do more experiments, we have added some experiments to verify the method. See P18.

We tried our best to improve the manuscript and made some changes in the manuscript. These changes will not influence the content and framework of the paper. And here we did not list the changes but marked in red in revised paper. We appreciate for Reviewers' warm work earnestly, and hope that the correction will meet with approval. Once again, thank you very much for your comments and suggestions.

Please also note the supplement to this comment:
https://www.nonlin-processes-geophys-discuss.net/npg-2017-49/npg-2017-49-AC2-supplement.pdf
* * *
$s(n)$

Seismic source

$b_1(n)$   $x_1(n)$        $x_2(n)$   $b_2(n)$

Sensor 1                    Sensor 2

**Fig. 1.**

**Fig. 2.**

(a) Gaussian signal

(a*) DSV corresponding to (a)

DSV-Dynamic Sample Variance

(b) α-stable distribution signal for (1.6,0,1,0)

(b*) DSV corresponding to (b)

(c) α-stable distribution signal for (1.2,0,1,0)

(c*) DSV corresponding to (c)

[Figure]

[Figure]

**Fig. 3.**

[Figure]

[Figure]

**Fig. 4.**

[Figure]

**Fig. 5.**

Interactive
comment

[Figure]

Fig. 6.

(a) α=0.8

(a*) Time delay estimation when α=0.8

(b) α=1.4

(b*) Time delay estimation when α=1.4

(c) α=1.8

(c*) Time delay estimation when α=1.8

(d) α=2.0

(d*) Time delay estimation when α=2.0

**Fig. 7.**

[Figure]

**Fig. 8.**

---

## Editor Comment (EC1) · S. Lovejoy (Editor) · 12 Mar 2018

Comments from editor (S. Lovejoy):

Beyond the seismology issues raised by the referees, there are a couple of other issues that I noticed.

a) The key variable  $\varphi$  is never given a name, nor explicitly related to probabilities. b) It is stated that: "the maximum difference between the number of positive and negative values is 92. Compared with the 3000 of sample data, this can be approximately considered as 0." This is a standard problem in nonparametric statistical testing. It is equivalent to asking what is the maximum excess of heads with respect to tails if a coin is tossed 3000 times. The authors' result (92) does not strike me as so negligible. But
even then if it is, a statistical test could only reject the hypothesis that the distribution was symmetric, not to accept it. The authors could use a standard maximum likelihood estimator for the parameters, thus giving standard errors for their parameter estimates. c) The authors do not show the evidence they used to estimate their key alpha values. This is such an important part of the paper that they should show a graph tending to support the validity of their technique. Alternatively, they could use the standard maximum likelihood method.

---

## Author Comment (AC3) · 3 Apr 2018

Q1: The key variable $\varphi$ is never given a name, nor explicitly related to probabilities. R1: Because there is no unified probability density function expression for $\alpha$-stable distribution, which only has a unified eigenfunction expression. The key variable $\varphi$ is a value of the eigenfunction. We added a description of $\varphi$ in the paper, please refer to P4L98. Thank you for your reminding.

Q2: It is stated that: "the maximum difference between the number of positive and negativevalues is 92. Compared with the 3000 of sample data, this can be approximately considered as 0." This is a standard problem in nonparametric statistical testing. It is equivalent to asking what is the maximum excess of heads with respect to tails if a coin

is tossed 3000 times. The authors' result (92) does not strike me as so negligible. But even then if it is, a statistical test could only reject the hypothesis that the distribution was symmetric, not to accept it. The authors could use a standard maximum likelihood estimator for the parameters, thus giving standard errors for their parameter estimates. R2: Thank you for your constructive suggestions. We have adjusted the research idea and proved the symmetry of microseismic signal by using maximum likelihood estimation. For detail, please refer to P6L155-P6L157. Accordingly, we modified the content of Fig. 5, please refer to Fig. 5 in P17.

Q3: The authors do not show the evidence they used to estimate their key alpha values. This is such an important part of the paper that they should show a graph tending to support the validity of their technique. Alternatively, they could use the standard maximum likelihood method. R3: It is really true as Reviewer suggested that the alpha values is a key parameter. Because of the complexity of microseismic signals, each microseismic signal corresponds to a different alpha value. We have discussed the estimation method of alpha in Section 3.2. Please refer to P7L187 – P8L214.

We tried our best to improve the manuscript and made some changes in the manuscript. These changes will not influence the content and framework of the paper. And here we did not list the changes but marked in red in revised paper. We appreciate for Editors' warm work earnestly, and hope that the correction will meet with approval. Once again, thank you very much for your comments and suggestions.

Please also note the supplement to this comment:
https://www.nonlin-processes-geophys-discuss.net/npg-2017-49/npg-2017-49-AC3-supplement.pdf

**Supplement:**

[revised manuscript text omitted]

---

## Author Response (AR3)

**Response to comments**

Q1: phi(t) is not an eigenfunction, rather it is the characteristic function of the probability density. This should be clearly stated. Also, it is confusing to use the variable "t" since it does not represent "time" (as it does elsewhere in the paper), but is rather the variable conjugate to the random variable.

A1: We are very sorry for our incorrect writing, and we have made correction according to the your comments. Please refer to P4L94 – P4L98.

Q2: Also, I know that you discussed your estimation method for alpha. I wanted a graph so that the reader could evaluate the evidence directly.

A2: Considering the your suggestion, we have add a Fig. 9 so that the reader could evaluate the evidence directly, and used estimate maximum likeihood estimator for alpha. Please refer to P11L276 – P11L282 and P20 Fig. 9

Special thanks to you for your good comments.

[revised manuscript text omitted]